# SPE–UPLC–MS/MS for Determination of 36 Monomers of Alkylphenol Ethoxylates in Tea

**DOI:** 10.3390/molecules28073216

**Published:** 2023-04-04

**Authors:** Qin Lin, Yujie Qin, Hezhi Sun, Xinru Wang, Mei Yang, Xinzhong Zhang, Li Zhou, Fengjian Luo

**Affiliations:** Tea Research Institute, Chinese Academy of Agricultural Sciences, Hangzhou 310008, China

**Keywords:** alkylphenol ethoxylates, SPE, tea, UPLC–MS/MS

## Abstract

Alkylphenol ethoxylates (APEOs) represent a non-ionic surfactant widely used as adjuvants in pesticide formulation, which is considered to cause an endocrine-disrupting effect. In the current study, we established a detection method for the APEOs residue in tea based on solid-phase extraction (SPE) for the simultaneous analysis of nonylphenol ethoxylates (NPEOs) and octylphenol ethoxylates (OPEOs) by UPLC–MS/MS. In the spiked concentrations from 0.024 to 125.38 μg/kg for 36 monomers of APEOs (*n*_EO_ = 3–20), the recoveries of APEOs range from 70.3–110.7% with RSD ≤ 16.9%, except for OPEO_20_ (61.8%) and NPEO_20_ (62.9%). The LODs of OPEOs and NPEOs are 0.008–2.09 μg/kg and 0.053–1.67 μg/kg, respectively. The LOQs of OPEOs and NPEOs are 0.024–6.27 and 0.16–5.01 μg/kg, respectively. OPEOs and NPEOs are detected in 50 marketed tea samples with a total concentration of 0.057–12.94 and 0.30–215.89 µg/kg, respectively. The detection rate and the range of the monomers of NPEOs are generally higher than those of OPEOs. The current study provides a theoretical basis for the rational use of APEOs as adjuvants in commercial pesticide production.

## 1. Introduction

Tea (*Camellia sinensis* L.) is the second most popular non-alcoholic beverage because of its special flavor and rich active ingredients [1], such as amino acids, flavonoids, alkaloids, tannins, phenols, and glycosides [2]. Pesticides are applied to control tea pests and diseases in order to guarantee the quality of tea and minimize crop loss. Pesticide adjuvants enhance the efficacy of active ingredients, and among them, surfactants represent the main category [3]. Customers’ focus is mostly on pesticide residues in tea, whereas the tea quality and safety related to adjuvant exposure do not attract significant attention. In fact, pesticide adjuvants account for 1–99% of pesticides, and they may cause a potential health threat to humans, through carcinogenicity, teratogenicity, mutagenicity, and endocrine disruption [3].

Alkylphenol ethoxylates (APEOs) are non-ionic surfactants widely applied to industrial products, such as pulp and paper, textiles, coatings, agricultural pesticides, lube oils, fuels, metals, and plastics due to their efficient and excellent biological profile, and about 60% of APEOs are discharged into the aquatic environment [4,5]. APEOs consist of a hydrophobic carbon chain and an ethylene oxide chain of 1–50 units, but typical products generally contain 9–10 units [6,7]. In addition, the chemical structures of APEOs are related and very similar, as presented in Appendix A. Nonylphenol (NP) and octylphenol (OP) are the degradation products of NPEOs and OPEOs in the environment, and they are more toxic and remain in the environment more persistently [7]. The distribution of APEOs and metabolites is very broad, and they are found in environmental media, such as sediments, water, paddy soil, and suspended and settling particles, and have been found in many countries, including Japan, the USA, Turkey, China, South Africa, Spain, Portugal, and other European countries [8,9,10,11,12,13,14,15,16]. In addition, APEOs were detected in food products, including seafood, pork, fruit juice, beehive samples, and vegetables [17,18,19,20,21]. APEOs and metabolites are known to be endocrine-disrupting chemicals (EDC) [22]. The negative effects on endocrine and reproductive systems are thought to be due to the ability of APEOs and their metabolites to: (1) mimic the effect of endogenous hormones, (2) antagonize the effect of endogenous hormones, (3) disrupt the synthesis and metabolism of endogenous hormones, and (4) disrupt the synthesis and metabolism of hormone receptors [23]. Therefore, the use of such compounds in production has been banned or strictly monitored in the European Union [24], and it is forbidden to use more than 0.1% content in biocide formulations there [24].

APEOs have superior surface properties and low cost, and they are used in commercial pesticide formulations to improve spray efficacy and increase the systematic movement of pesticides in plants and animals [8,19]. In agriculture, frequently spraying pesticides containing surfactants and applying sewage and sludge as fertilizers may result in APEOs contamination of crops [24,25,26,27]. APEOs may accumulate in the human body via food containing APEOs. It was reported that APEOs and their metabolites were found in the human subcutaneous adipose tissue ranging from 6 to 80 ng/g Fw; moreover, OP (0.08 ng/mL), OPEO_1_ (0.07 ng/mL), and OPEO_2_ (0.16 ng/mL) were detected in human breast milk [28,29]. Metabolites of APEOs were also traced in urine [30].

Visual chromatograms achieve the purpose of monitoring APEOs [31]. Gas chromatography (GC) and liquid chromatography (LC) are usually coupled to mass spectrometry to analyze APEOs [29]. Nevertheless, when APEOs contain more than five ethoxylation units, their volatility and thermal stability become poor, requiring derivatization for GC–MS analysis. LC–MS/MS was developed to analyze APEOs, providing better selectivity and lower detection limits [32]. Matrices of leaf vegetables and beehive samples were successfully analyzed by LC–MS/MS [18,19]. NPEO_2–20_ and OPEO_2–20_ in leafy vegetables (cabbage, lettuce, and spinach) were determined with recoveries of 72.8–122.6% and LOQs of 0.18–1.75 µg/kg [18]. NPEO_3–13_ and OPEO_3–13_ in beehive matrices (honey, pollen, and wax) were measured, with recoveries from 75 to 111% [19].

In this study, we developed a method for analyzing APEOs with between 3 and 20 ethoxylation units (*n*_EO_ = 3–20) in a tea matrix, and the investigated residue provides a theoretical basis for further risk assessment of the intake of APEOs via tea consumption. Moreover, the current study provides proper guidance for the rational use of APEOs as adjuvants in pesticide production.

## 2. Results and Discussion

### 2.1. Distributions of NPEOs and OPEOs Oligomers

APEOs standards were accurately weighed and dissolved in acetonitrile with 200 μg/mL and further diluted to use. It is difficult to obtain the monomer of APEOs, and the monomer in the available standard does not have clear purity. For quantitative analysis, we need to clarify the proportional distribution of monomers in the standard [33]. The mass response of all monomers of NPEOs or OPEOs was obtained by using MS, and the mass response of homologs as a whole was added; the ratio of the monomer response to the total response, representing the ratio of the monomer content in the standard, was calculated by the formulas in Section 3.5. As represented in Figure 1, the proportion of each monomer in standard solutions for OPEOs and NPEOs ranges from 0.05 to 12.54% and from 0.27 to 10.02%, respectively. This method of conversion to percentage is normalized [23,34]. Quantitatively calculated oligomers of NPEOs and OPEOs in the sample are based on these alkylphenol ethoxylate standards.

### 2.2. UPLC–MS/MS Conditions Optimization

The MRM method produces characteristic fragment ions by optimizing cone voltage and collision energy in the positive mode with the ESI source. Based on the relationship between MS–MS ions, the fragmentation path for the collision-induced dissociation (CID) method was used to quantitatively generate daughter ions, as shown in Appendix A. Selecting an ammonium acetate solution as a mobile phase was beneficial for the ionization of monomers of OPEOs and NPEOs by adding ammonium ions and forming [M + NH_4_]^+^ or [M + 2NH_4_]^2+^ ions, which was monitored using mass spectrometry. Appendix A shows the results of full-scan mass spectra of OPEOs and NPEOs. The mass spectrum for a series of monomers exhibits a difference of 44 or 22 mass units. Monomers with ethoxylation units in OPEO (9–10), OPEO (16), NPEO (4), NPEO (9–10), and NPEO (14) are 3–15, 9–20, 3–10, 3–15, and 6–20, respectively. When the number of ethoxylation units is at least 13 (*n* ≥ 13), [M + NH_4_]^+^ and [M + 2NH_4_]^2+^ ions simultaneously exist, while at *n* ≥ 15, the ionic abundance of [M + 2NH_4_]^2+^ is close to or exceeds the ionic abundance of [M + NH_4_]^+^. Finally, OPEO_16–20_ and NPEO_16–20_ binding to two ammonium ions were selected, except NPEO_17_ and NPEO_19_ due to interfering ions near these two ions, as shown in Appendix A. The monitored ammonium adducts of APEOs are shown in Appendix A. The retention time of monomers of OPEOs and NPEOs ranges from 6.11 to 5.97 min and 6.34 to 6.77 min, respectively. The retention time decreases with the number of ethoxylation units.

### 2.3. Optimization of Extraction and Clean-Up

The tea matrix is so complex that many active substances are extracted, which may interfere with the analysis of target compounds [35]. Optimizing the extraction and purification methods may help effectively reduce the interference with other components and improve the extraction efficiency of the target. In this study, Oasis HLB was selected to filter impurities and enrich target components. The Oasis HLB absorbent is a macroporous copolymer composed of hydrophilic N-vinylpyrrolidone and lipophilic divinylbenzene; the π–π ring of benzene can interact with the phenyl group of the surfactant. This polymeric phase exhibits great loading capacity, and its chemical properties are stable when drying the column [33,36]. Considering the properties of APEOs, the HLB column is suitable for extracting APEOs [36,37,38].

A standard solution of APEOs was added to a blank tea sample in three duplicates, several groups of different variables were set at a certain stage in the extraction and clean-up process, and the best treatment group was selected by comparing the recovery rate and RSD. The results show that the components of the leaching solution greatly influence the results. Compared with the methanol–water mixed solution containing 1%, 3%, 5%, and 10% NH_3_‧H_2_O with recoveries of 86–121%, 71–103%, 57–91%, and 55–102%, respectively, the solution containing 1% NH_3_‧H_2_O exhibits the highest recovery (Figure 2). The methanol–water volume ratio in the leaching solution is optimized (Figure 3). The recoveries of APEOs of methanol–water volume ratio with 4:6 (*v*/*v*), 5:5 (*v*/*v*), 6:4 (*v*/*v*), and 7:3 (*v*/*v*) are 79–112%, 86–109%, 65–92%, and 19–66%, respectively. When the methanol–water volume ratio is 4:6 (*v*/*v*) and 5:5 (*v*/*v*), the recovery rate is higher than 6:4 (*v*/*v*) and 7:3 (*v*/*v*), which may be caused by the poor interaction between HLB and APEOs at a methanol–water volume ratio of 6:4 (*v*/*v*) and 7:3 (*v*/*v*); hence, the target compounds are washed out when the sample passes through cartridges [6]. Considering the cost, we chose the methanol–water 4:6 (*v*/*v*) mixed solution. The content of dichloromethane in the methanol for the elution was set at 5%, 25%, 50%, and 80%, and the recoveries of APEOs were 80–90%, 75–90%, 80–90%, and 80–110%, respectively (Figure 4a,b). The mixture of methanol and dichloromethane alters the polarity of the solution [6]. The elution volume is 2, 3, 4, and 5 mL, and the recoveries all range from 80% to 100%, and RSD% is less than 15% (Figure 5a,b). Using smaller amounts of organic solvents effectively saves experimental time and cost. Therefore, 2 mL of methanol containing 5% dichloromethane was chosen for the elution.

### 2.4. Method Validation

Monomers of APEOs were measured using HPLC–MS/MS (Appendix A). The validation of analytical methods was carried out using parameters linearity, the coefficient of determination (R^2^), recoveries, relative standards (RSDs), the limit of detection (LOD) and the limit of quantitation (LOQ). Tea samples without target components were prepared for matrix-matching standard curves to evaluate the linearity and R^2^. Blank tea samples were spiked with three levels and repeated five times to calculate recoveries and RSDs, determining precision and accuracy. The lowest spiked concentrations were treated as LOQ.

The precision and accuracy were verified by measuring recoveries of spiked OPEOs and NPEOs in tea. The validation parameters of the developed method are listed in Appendix A. The three levels of spiked concentrations of APEO_3–20_ in tea range from 0.024 to 125.38 μg/kg. Except for OPEO_20_ and NPEO_20_, the recoveries of the rest monomers are more than 70%. The R^2^ ranges from 0.9949 to 0.9999 for OPEOs with RSDs 1.5–14.5%, and from 0.9978 to 0.9999 for NPEOs with RSDs 2.0–16.9%. The LODs of OPEOs and NPEOs are 0.008–2.09 μg/kg and 0.053–1.67 μg/kg, respectively. The LOQs of OPEOs and NPEOs is 0.024–6.27 μg/kg and 0.16–5.01 μg/kg, respectively. The selectivity and sensitivity of the method were satisfying for detecting APEOs.

As described in Table 1, the present method has obvious advantages. Sediments were extracted to analyze APEO_3–12_ using the Oasis HLB column, and the results show that recoveries of APEO_6–12_ are 78–92%, but APEO_3–5_ is merely 60% [39]. Absolute extraction recoveries of APEO_2–20_ in the water sample via the SPE method combined with LC–MS/MS range from 37 to 69% [34]. In addition, many research reports mainly focused on the number of ethoxylation units of 15 or less (*n*_EO_ ≤ 15) [19,36,40,41,42]. The matrix is more complex in this study, and the number of APEOs monomers is larger than in previous studies [34,36,39,40,41,42].

### 2.5. Residues in Marketed Tea Samples

APEO_3–20_ in marketed tea samples was determined, and the concentration levels are illustrated in Figure 6. The residues of OPEOs monomers range from 0.003 to 1.94 µg/kg, but OPEO_14–20_ is not detected. The total concentration of OPEOs is between 0.057 and 12.94 µg/kg. The monomers with the highest detection concentration are OPEO_9_, OPEO_10_, and OPEO_11_. These values match the detection results of OPEOs in the leafy vegetable samples. The sum residues of OPEO_2–20_ in leaf vegetables was 0–8.67 µg/kg, and the highest detection monomers were OPEO_10_, and OPEO_11_ [18], while it was up to 398 µg/kg in beeswax [19]. The detection frequencies are between 6.0% and 40.0% (average 25.2%), and OPEO_3–4_ and OPEO_7–11_ account for more than 28.0%, among which, OPEO_9_ has the highest frequency. OPEOs are detected in 21 tea samples, but NPEOs are detected in 49 samples. Almost all tea samples are contaminated by APEOs, and APEOs were detected in all vegetable samples and beehive samples [18,19]. Considering other parameters, including concentration, detection frequency, and the number of detected monomers, there are more NEPOs than OPEOs. The total determined concentration of NPEOs is from 0.30 to 215.89 µg/kg, and the concentration of each monomer is from 0.009 to 37.26 µg/kg. NPEO_3–19_ is detected with detection frequencies of 2.0–98.0% (average 55.1%), and nine components, i.e., NPEO_3–4_ and NPEO_6–12_, account for more than 70%. The total concentration of NPEOs in tea is higher than in leaf vegetables and honey, but much lower than in beeswax [18,19]. In market tea samples, the concentration and detection rate of NPEOs are higher than for OPEOs. The total concentration of NPEOs was higher than that of OPEOs in market vegetables, which might also originate from the more frequent use of NPEOs in production than OPEOs [18].

## 3. Materials and Methods

### 3.1. Reagents, Chemicals, and Materials

OPEO (9–10), OPEO (16), NPEO (4), NPEO (9–10), and NPEO (14) were purchased from Beijing Zhenxiang Technology Co., Ltd (Beijing, China). These five standards were all composed of homologous monomers. Two adjacent monomers in the same standard differed by one ethoxylation unit. The standards were expressed by NPEO (X) or OPEO (X), where (X) represented the average number of ethoxylation units. The Oasis HLB cartridge (3 cc, 60 mg) was purchased from Waters Corporation (Milford, MA, USA). LC–MS grade acetonitrile and methanol were provided by Merck KGaA. LC–MS (Darmstadt, Germany) grade ammonium acetate was purchased from ANPEL Technologies (Shanghai) Inc (Shanghai, China). Analytic grade methanol and dichloromethane were purchased from Jiangsu Yonghua Chemical Reagent. Analytic grade NaCl was purchased from Guangzhou Jinhua Chemical Regent CO. (Jinhua, China), Ltd. Standard stock solutions of OPEO (9–10), OPEO (16), NPEO (4), NPEO (9–10), and NPEO (14) with a total concentration of 2.00 × 10^5^ µg/L were prepared by accurately weighing the standard compounds and dissolving in LC–MS grade acetonitrile, respectively, and they were stored in the dark at −20 °C. Working standard solutions were obtained with LC–MS grade methanol by further dilution of all stock solutions with concentrations of 1.00 × 10^4^, 5.00 × 10^3^, 1.00 × 10^3^, 500, 100, 50.0, and 10.0 µg/L.

### 3.2. Blank Tea Samples

The production of green tea accounts for over 60% of the total tea production in China [43]. Therefore, we chose green tea as a representative for method validation. The blank tea sample was made at an organic tea factory which meant no pesticide formulations had been used in the tea production, and it was processed according to the extraction and the instrumental analysis method described in Section 3.3 and Section 3.4 to ensure that it was not contaminated or that its response in mass spectra was negligible. The blank tea sample was ground into powder and stored separately in the dark at −20 °C. In the experiment, the treatment conditions of the blank sample was completely consistent with other detected samples, and no additional reagent was added.

### 3.3. Sample Collection and Preparation

A total of 50 tea samples were randomly collected from the market. The tea samples were pulverized with a stainless-steel grinder and then frozen for preservation at −20 °C for further analysis.

In the current study, the process of extraction was based on a modified SPE method, and the method derived from the Chinese national standard for the determination of APEOs in textiles [44]. To start, 2 g of each tea sample was soaked in 2 mL of pure water and 8 mL of methanol for 30 min. After vortexing for 1 min and centrifugation at 8000 rpm for 5 min, 2 mL of supernatant was transferred into a 10 mL centrifuge tube and thoroughly mixed with 2 mL of water. Then, it was immediately transferred to the Oasis HLB column pre-activated with 2 mL of methanol and 4 mL of water, and then 8 mL of methanol and water with a volume ratio of 6:4 containing 1% NH_3_‧H_2_O was added through the HLB column. Then, the column was drained and eluted with 2 mL methanol containing 5% dichloromethane. The flow rate was controlled at 1 mL/min. Eluent was collected and rotary-evaporated completely. Finally, the residue was dissolved with 1 mL methanol and filtered into vials for further analysis by UPLC–MS/MS.

### 3.4. Instrumental Analysis

APEOs were analyzed by UPLC–MS/MS using Waters ACQUITY UPLC**^®^** H-Class equipped with Waters Xevo**^®^** TQ-S Micro (Waters Corporation, Milford, MA, USA ), and the system was run by Masslynx 4.1 software. The chromatograph was performed for installation of ACQUITY UPLC**^®^** BEH C18 chromatographic column (1.7 µm 2.1 × 100 mm), and the column oven temperature was set at 40 °C. The flow rate was maintained at 0.25 mL/min. The mobile phases A and B were 2 mmol ammonium acetate in water and acetonitrile, respectively. The solvent gradient was set as follows: 0–1 min, 60–50% A; 1–4 min, 50–15% A; 4–8 min, 15–1% A; 8–8.7 min, 1–1% A; and 8.7–9.2 min, 1–60% A. An ESI probe in the positive ion mode with the multiple reaction monitoring (MRM) mode was used in the mass spectrometer. The parameters for detecting target components were set as follows: capillary voltage: 0.2 kV; degassing temperature: 350 °C; degassing flow rate: 650 L/h; cone follow rate: 50 L/h; ion source temperature: 150 °C; injection volume: 5 μL.

### 3.5. Date Analysis

All data analyses were processed using Microsoft Excel and Origin 2019.

Individual monomers of NPEOs and OPEOs in samples were evaluated by the following formulas, which referred to the report of DeArmond et al. [34].
(1)
$$X=\frac{A_{x}}{A_{s}}\times C_{s}\times p_{n}\times V$$

(2)
$$p_{n}=\frac{\sum a_{n}}{\sum V_{n}}$$

(3)
$$a_{n}=\frac{\frac{A_{n}}{\sum A_{n}}\times M_{n}}{\sum (M_{n}\times \frac{A_{n}}{\sum A_{n}})}$$


In the formulas, *X* is the residual concentration of the monomer of OPEOs and NPEOs in samples. *A*_x_ and *A*_s_ represent the peak area of samples and working standard solutions, respectively. *V* denotes the samples’ volume. *p_n_* stands for the ratio of each monomer after mixing two or three homologous standards. V*_n_* is the corresponding constant volume of working standard solutions. *a_n_* indicates the percentage distribution of the monomer in one of the standards. *A_n_* is the peak area of the monomer in working standard solutions, and *M_n_* is the corresponding relative molecular mass.

## 4. Conclusions

In this paper, we established a SPE–UPLC–MS/MS analytical method to determine APEO_3–20_ (total 36 monomers) residues. The selectivity and sensitivity of the method were satisfying for detecting APEOs in tea. The linearity is in the detection range, with an R^2^ of 0.9949–0.9999. The recoveries of APEO_3–20_ are 61.8–110.7%, with RSD 1.5–16.9%, and the range of LOD is 0.008–2.09 μg/kg, and LOQ is 0.024–6.27 μg/kg. A total of 50 market tea samples were collected to determine APEOs, and 49 samples were contaminated APEOs by this method. OPEO_3–13_ and NPEO_3–19_ are detected with a total concentration of 0.057–12.9 µg/kg and 0.30–215.89 µg/kg, respectively. The detected rate of OPEO_3–13_ and NPEO_3–19_ is 6.0–40.0% (average 25.2%) and 2.0–98.0% (average 55.1%), respectively. The number of detectable monomers of NPEOs (*n*_EO_ = 3–19) is higher than OPEOs (*n*_EO_ = 3–13). Metabolites of APEOs, including OPEO_1–2_, NPEO_1–2_, NP, and OP, must be carefully treated in marketed tea samples. The risk assessment of dietary intake in tea soups should also be further determined.

## Figures and Tables

**Figure 1 molecules-28-03216-f001:**
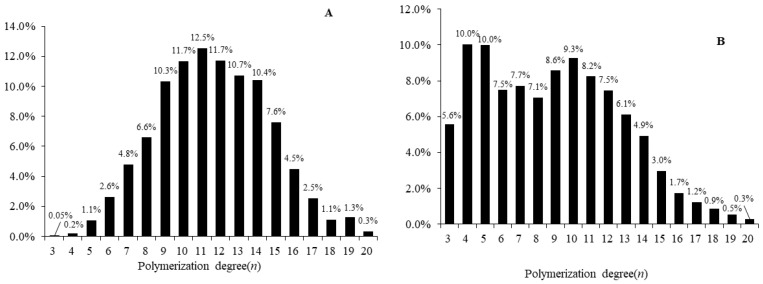
The proportion of each monomer in (**A**) OPEO_3–20_ and (**B**) NPEO_3–20_.

**Figure 2 molecules-28-03216-f002:**
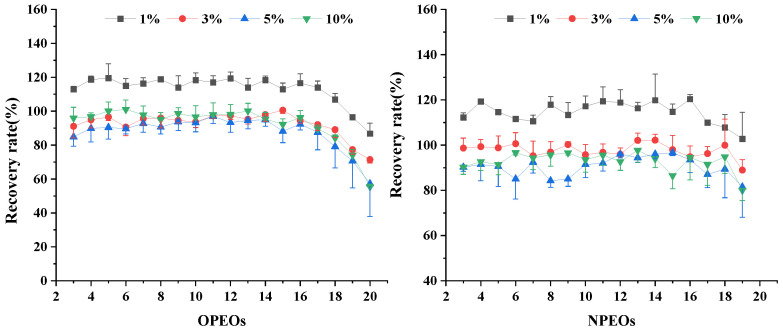
Effect of the ammonia content in the leaching solvent on APEOs.

**Figure 3 molecules-28-03216-f003:**
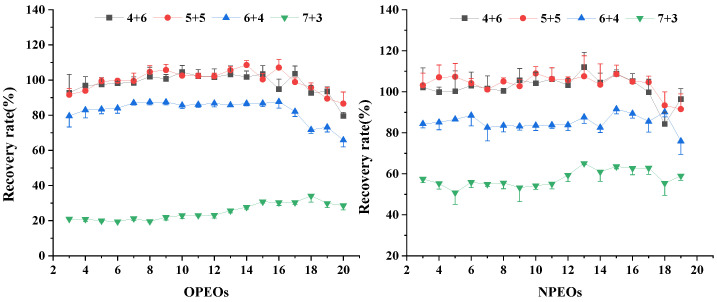
Effect of the methanol–water volume ratio in the leaching solvent on the recoveries of APEOs.

**Figure 4 molecules-28-03216-f004:**
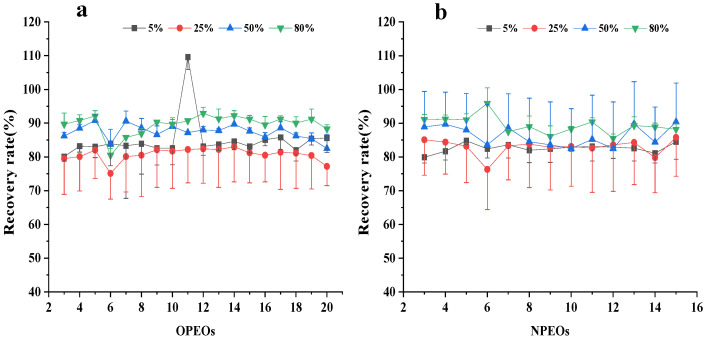
Effect of the dichloromethane content in HLB SPE column elution solution on OPEOs (**a**) and NPEOs (**b**) recoveries.

**Figure 5 molecules-28-03216-f005:**
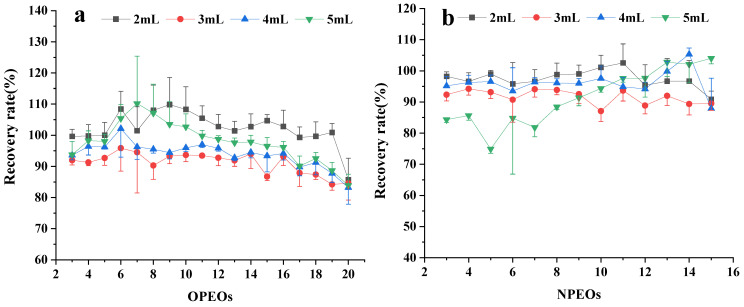
Effect of HLB SPE column elution solution volume on OPEOs (**a**) and NPEOs (**b**) recoveries.

**Figure 6 molecules-28-03216-f006:**
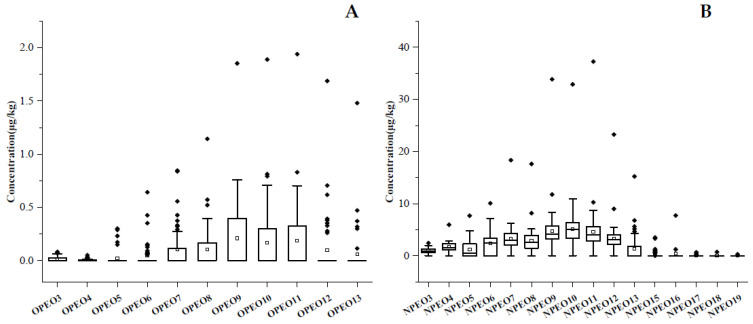
Concentrations of (**A**) OPEOs and (**B**) NPEOs in marketed tea samples.

**Table 1 molecules-28-03216-t001:** Comparison of the present method with other reported methods on APEO analysis.

Analyte	Matrix	Enrichment	Optimisation of Extraction	Detection	Recovery (%)	Ref.
APEO_3–13_	Beehive samples	QuEChERS	–	LC–MS	74–111	[19]
OPEO_7–11_	Water	SPE	Cartridges	LC–MS/MS	58.7–68.4 (absolute recovery)	[36]
APEO_2–20_	Water	SPE	Cartridges Elution solvents	HPLC–MS/MS	37–69 (absolute recovery)	[34]
APEO_1–12_	Sewage	SPE	Cartridges pH and ionic strength Sample volume Wash step Elution solvents	LC–MS/MS	60–108	[39]
APEO_1–15_	Soil sludge	PLE *–SPE	Solvents	LC–APCI–MS	89–102	[40]
APEO_1–8_	Sewage	SPE	Sample volume pH Wash step Elution solvents	LC–MS/MS	74–106	[41]
APEO_1–8_	Wastewater	On-line SPE	Sorbents Loading volume Elution solvents	LC–MS/MS	50–120	[42]
APEO_3–20_	Tea	SPE	Elution solvents Wash step Elution solvents volume	LC–MS/MS	70.3–110.7 (APEO_20_ 61.8%–62.9%)	This work

* PLE: pressurized liquid extraction.

## Data Availability

All data are included in this article.

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
