# Peer review of "SPE–UPLC–MS/MS for Determination of 36 Monomers of Alkylphenol Ethoxylates in Tea"

_molecules, 2023, doi:10.3390/molecules28073216_

Round 1

Reviewer 1 Report

Reviewer’s Comments:

The manuscript “Analysis of Alkylphenol Ethoxylates in Tea by SPE Coupled to UPLC-MS/MS” is a very interesting work. This work describes the Alkylphenol ethoxylates (APEOs) is a kind of widely used non-ionic surfactant, as an adjuvant in pesticide formulation, which is considered to cause endocrine disrupting effect. In the current study, detection method for APEOs residue in tea was established, on the basis of SPE (sol-id-phase extraction) for simultaneous analysis of nonylphenol ethoxylates (NPEOs) and oc-tylphenol ethoxylates (OPEOs) by UPLC-MS/MS. In the spiked concentrations from 0.024 μg/kg to 125.38 μg/kg for 36 monomers of APEOs (nEO = 3-20), the recoveries of APEOs ranged from 70.3-110.7% with RSD ≤16.9%, except OPEO20 (61.8%) and NPEO20 (62.9%). The results are consistent with the data and figures presented in the manuscript. While I believe this topic is of great interest to our readers, I think it needs major revision before it is ready for publication. So, I recommend this manuscript for publication with major revisions.

1. In this manuscript, the authors did not explain the importance of the Alkylphenol Ethoxylates in the introduction part. The authors should explain the importance of Alkylphenol Ethoxylates.

2) Title: The title of the manuscript is not impressive. It should be modified or rewritten it.

3) Correct the following statement “The detection rate and the range of monomers of NPEOs was generally higher than that of OPEOs. The current study would provide a theoretical basis to guide the ra-tional use of APEOs as adjuvant in commercial pesticide production”.

4) Keywords: There so many keywords and reduce them up to 5. So, modify the keywords.

5) Introduction part is not impressive. The references cited are very old. So, Improve it with some latest literature like 10.3390/biom12010083, 10.3390/ph15101164

6) The authors should explain the following statement with recent references, “The total concentration of OPEOs was between 0.057 µg/kg and 12.9 µg/kg. The monomers with the highest detection concentration were OPEO9, OPEO10, and OPEO11”.

7) Add space between magnitude and unit. For example, in synthesis “21.96g” should be 21.96 g. Make the corrections throughout the manuscript regarding values and units.

8) The author should provide reason about this statement “50 tea samples in total were randomly collected from the market, including 24 white tea, 12 black tea, 5 dark tea, 4 green tea, 3 scented tea, and 3 oolong tea”.

9) Comparison of the present results with other similar findings in the literature should be discussed in more detail. This is necessary in order to place this work together with other work in the field and to give more credibility to the present results.

10) Conclusion part is very long. Make it brief and improve by adding the results of your studies.

11) There are many grammatic mistakes. Improve the English grammar of the manuscript.

Author Response

Dear reviewer,

Thank you for your valuable comments concerning our manuscript entitlied “Analysis of Alkylphenol Ethoxylates in Tea by SPE Coupled to UPLC-MS/MS” (Manuscript ID: 2281710). Those comments are all helpful for revising and improving our paper. We have studied all comments carefully and have made conscientious correction. The main correction in the paper and the responds to the reviewer’s comments are as following.

------------------------------------------------------------

The manuscript “Analysis of Alkylphenol Ethoxylates in Tea by SPE Coupled to UPLC-MS/MS” is a very interesting work. This work describes the Alkylphenol ethoxylates (APEOs) is a kind of widely used non-ionic surfactant, as an adjuvant in pesticide formulation, which is considered to cause endocrine disrupting effect. In the current study, detection method for APEOs residue in tea was established, on the basis of SPE (sol-id-phase extraction) for simultaneous analysis of nonylphenol ethoxylates (NPEOs) and oc-tylphenol ethoxylates (OPEOs) by UPLC-MS/MS. In the spiked concentrations from 0.024 μg/kg to 125.38 μg/kg for 36 monomers of APEOs (nEO = 3-20), the recoveries of APEOs ranged from 70.3-110.7% with RSD ≤16.9%, except OPEO20 (61.8%) and NPEO20 (62.9%).The results are consistent with the data and figures presented in the manuscript. While I believe this topic is of great interest to our readers, I think it needs major revision before it is ready for publication. So, I recommend this manuscript for publication with major revisions.

  1. In this manuscript, the authors did not explain the importance of the Alkylphenol Ethoxylates in the introduction part. The authors should explain the importance of Alkylphenol Ethoxylates.

Response:

Thank you for this valuable feedback. We have added the content from Line 53 to Line 55 to explain the importance of Alkylphenol Ethoxylates.

2) Title: The title of the manuscript is not impressive. It should be modified or rewritten it.

Response:

We appreciate the reviewer for this kind recommendation. The paper title is modified from “Analysis of Alkylphenol Ethoxylates in Tea by SPE Coupled to UPLC-MS/MS” to SPE-UPLC-MS/MS for Determination of 36 Monomers of Alkylphenol Ethoxylates in Tea”.

3) Correct the following statement “The detection rate and the range of monomers of NPEOs was generally higher than that of OPEOs. The current study would provide a theoretical basis to guide the ra-tional use of APEOs as adjuvant in commercial pesticide production”.

Response:

We regret there were problems with the English. We have made a correction to “The detection rate and the range of the monomers of NPEOs were generally higher than those of OPEOs. The current study provides a theoretical basis for the rational use of APEOs as adjuvants in commercial pesticide production.” (Line 17-19)

4) Keywords: There so many keywords and reduce them up to 5. So, modify the keywords.

Response:

We thank the reviewer for raising this question. We delete “Pesticide adjuvants”, and four keywords are reserve--Alkylphenol Ethoxylates; SPE; Tea; UPLC-MS/MS.

5) Introduction part is not impressive. The references cited are very old. So, Improve it with some latest literature like 10.3390/biom12010083, 10.3390/ph15101164

Response:

Thank you for this valuable feedback. We have added new content in Introduction, which includes Line 24-25, 30-32, and 51-55 and 63, to enrich the Introduction. We have improved some newest articles to replace the original one, which is [2], [3], [6], [15], [17], [23], [26], [27], [32], and [33]. And [3] and [33] are recommended by reviewer.

6) The authors should explain the following statement with recent references, “The total concentration of OPEOs was between 0.057 µg/kg and 12.9 µg/kg. The monomers with the highest detection concentration were OPEO9, OPEO10, and OPEO11”.

Response:

We thank for the reviewer recommendation. We have added sentences to explain the result. (Line 193-196).

7) Add space between magnitude and unit. For example, in synthesis “21.96g” should be 21.96 g. Make the corrections throughout the manuscript regarding values and units.

Response:

We are so graceful for your kind question. We have repeatedly checked the article to correct this type of error.

8) The author should provide reason about this statement “50 tea samples in total were randomly collected from the market, including 24 white tea, 12 black tea, 5 dark tea, 4 green tea, 3 scented tea, and 3 oolong tea”.

Response:

Thank you for your question. Tea samples are not intended to be purchased for specific types and sample quantities. After classification of tea samples, the small quantity is not convincing. And the relationship between tea types and APEOs residues is not analyzed in paper. Hence, we decide to delete the information on the quantity of different types of tea, and combined sample collection and extraction and clean-up into a new section. (Section 4.3, page 9-10).

9) Comparison of the present results with other similar findings in the literature should be discussed in more detail. This is necessary in order to place this work together with other work in the field and to give more credibility to the present results.

Response:

Thank you for pointing that out. In addition to adding Line 193-196, it also adds Line 199-200, and Line 205-207, citing the results of articles on similar topics, making the data in this article more convincing. (Page 8)

10) Conclusion part is very long. Make it brief and improve by adding the results of your studies.

Response:

Thank you for this valuable feedback. We have merged sentences that were semantically similar, and removed those that were not necessary in the conclusion. More details of data are presented. (5. Conclusions, page 11)

11) There are many grammatic mistakes. Improve the English grammar of the manuscript.

Response:

We regret there were problems with the English. The paper has been carefully revised by a professional language editing service to improve the grammar and readability.

Reviewer 2 Report

In its current form, the manuscript can be accepted.

Author Response

Dear reviewer,

Thank you very much for your review of our manuscript entitlied “Analysis of Alkylphenol Ethoxylates in Tea by SPE Coupled to UPLC-MS/MS” (Manuscript ID: 2281710). 

Reviewer 3 Report

The manuscript “Analysis of Alkylphenol Ethoxylates in Tea by SPE Coupled to UPLC-MS/MS” by Lin Qin, Qin Yujie, Sun Hezhi, Wang Xinru, Yang Mei, Zhang Xinzhong, Zhou Li, Luo Fengjian describes the optimization of some parameters of the method for determination of alkylphenol ethoxylates in tea samples using UPLC-MS/MS. I think that due to their toxic properties, the determination of these compounds in various types of teas is advisable. Therefore, a rapid method for their determination is needed.

I have some comments which could influence on the value of the manuscript and can improve their quality:

 1. Is Germany not part of the European Union? Line 48 page 2.

2.       Not all tables and figures need to be included in the main manuscript. Some can be moved to supplementary material (for example table 1 and table 2 or figure 1, 2 and 8).

3.       Figure 3 is unreadable, maybe from all figures make two for example: a) OPEO, b) NPEO.

4.       Whether spaces are needed after the colon? line18-23 page 5

5.       Shouldn't the stoichiometric indices in the formulas be in subscript? Figure 5 and 6

6.       Was the validation of the method carried out according to some guide? If yes, with what document. If not, what are the acceptable validation parameters and from where they were taken?

7.       The results and discussion include some elements of comparison with another methods, but not enough. Authors should describe advantages of the method compared to other reported methods. Can prepare a table with a short description of methods optimized by other researchers.

8.       I think that figure 8 „Concentration of…” should be figure 9. Please, change the numer of figure in the text and under the figure.

9.       Different types of tea were studied, so the following question arises: was there any correlation noted for the occurrence of any of the tested compounds in a particular type of tea? (section 2.4)

10.    What type of tea was used for validation and why?

11.    I know that the authors plan to assessment of dietary intake in the future. However, my question is why not do it in this work. It would increase the value of this work.

12.    Preparing references is not in accordance with the requirements of the journal. Titles of articles should be abbreviated and capitalized: Author 1, A.B.; Author 2, C.D. Title of the article. Abbreviated Journal Name Year, Volume, page range.

13.    Please, corrected the word „china” line 106 page 15; the abbreviation „quechers” line 71 page 17 etc. Please, check all position in references very carefully.

Author Response

Dear reviewer,

Thank you for your valuable comments concerning our manuscript entitlied “Analysis of Alkylphenol Ethoxylates in Tea by SPE Coupled to UPLC-MS/MS” (Manuscript ID: 2281710). Those comments are all helpful for revising and improving our paper. We have studied all comments carefully and have made conscientious correction. The main correction in the paper and the responds to the reviewer’s comments are as following.

------------------------------------------------------------

The manuscript “Analysis of Alkylphenol Ethoxylates in Tea by SPE Coupled to UPLC-MS/MS” by Lin Qin, Qin Yujie, Sun Hezhi, Wang Xinru, Yang Mei, Zhang Xinzhong, Zhou Li, Luo Fengjian describes the optimization of some parameters of the method for determination of alkylphenol ethoxylates in tea samples using UPLC-MS/MS. I think that due to their toxic properties, the determination of these compounds in various types of teas is advisable. Therefore, a rapid method for their determination is needed.

  1. I have some comments which could influence on the value of the manuscript and can improve their quality:
  1. Is Germany not part of the European Union? Line 48 page 2.

Response:

Thank you for this valuable feedback. We changed the sentence "Therefore, the use of such compounds in production has been banned or strictly monitored in Germany, Switzerland and European Union." into "Therefore, the use of such compounds in production has been banned or strictly monitored in the European Union " (Line 51, page 2)

  1. Not all tables and figures need to be included in the main manuscript. Some can be moved to supplementary material (for example table 1 and table 2 or figure 1, 2 and 8).

Response:

We appreciate the reviewer for this kind recommendation. We have added supplementary materials, including the following contents.

Table 1 Mass spectrometric parameters for monitoring OPEO3-20 and NPEO3-20. (Now Table S1)

Table 2 Validated parameters of APEO3-20 in spiked tea samples. (Now Table S2)

Figure 1. Full-scan mass spectra of (A) OPEO (9-10) (nEO = 3-15), (B) OPEO (16) (nEO = 9-20), (C) NPEO (4) (nEO = 9-11), (D) NPEO (9-10) (nEO = 3-15), and (E) NPEO (14) (nEO = 6-20) in 5 mg/kg standards. (Now Figure S1)

Figure 2. Chromatograms of (A) NPEO17 and NPEO19 combined two ammonium ions, and (B) NPEO17 and NPEO19 combined one ammonium ion. (Now Figure S2)

Figure 8. Multiple reaction-monitoring chromatograms of HPLC-MS/MS for blank tea samples spiked at the middle concentration of OPEOs and NPEOs. (Now Figure S2)

  1. Figure 3 is unreadable, maybe from all figures make two for example: a) OPEO, b) NPEO.

Response:

We thank the reviewer for raising this question. We have added detailed data to describe the Figure 3. (Now Figure 1). We also neglected the description of Figure 4 (now Figure 2) and have now added the data description. (Line 116, 118-120, page 3)

  1. Whether spaces are needed after the colon? line18-23 page 5

Response:

We are so graceful for your kind question. We have deleted the spaces after the colon. (Line 119-125, page 3)

  1. Shouldn't the stoichiometric indices in the formulas be in subscript? Figure 5 and 6

Response:

Thank you for your question. We have improved the Figure 5 (Now Figure 3) and Figure 6 (Now Figure 4). We tried to subscript the numbers in the formula, but that would make the numbers too small to be clear, so we simply wrote the numbers of the ethoxylation units and labeled the compound categories in each diagram. (Page 6)

  1. Was the validation of the method carried out according to some guide? If yes, with what document. If not, what are the acceptable validation parameters and from where they were taken?

Response:

We thank the reviewer for raising this question. Our method carried out according to a Chinese national standard. We have added a sentence to clarify this. (Line 243-245, page 10)

  1. The results and discussion include some elements of comparison with another methods, but not enough. Authors should describe advantages of the method compared to other reported methods. Can prepare a table with a short description of methods optimized by other researchers.

Response:

We thank for the reviewer recommendation. We have added a table (Table 1) to compare with reported methods. (Page 7-8)

  1. I think that figure 8 „Concentration of…” should be figure 9. Please, change the numer of figure in the text and under the figure.

Response:

Thank you for your kind question. We have corrected it (Now Figure 6). (Line 190, page 8)

  1. Different types of tea were studied, so the following question arises: was there any correlation noted for the occurrence of any of the tested compounds in a particular type of tea? (section 2.4)

Response:

We thank the reviewer for raising this question. After classification of tea samples, the small quantity is not convincing. And the relationship between tea types and APEOs residues is not analyzed in paper. Hence, we decide to delete the information on the quantity of different types of tea and combined sample collection and extraction and clean-up. (Section 4.3, page 9-10).

  1. What type of tea was used for validation and why?

Response:

Thank you for this valuable feedback. We choose green tea for validation, because it is the most productive tea in China. We have added sentences in details to clarify these questions in Section 4.2. (Line 231-238, page 10)

  1. I know that the authors plan to assessment of dietary intake in the future. However, my question is why not do it in this work. It would increase the value of this work.

Response:

We regret that risk assessment of dietary intake is not covered in the article. If a dietary risk assessment is carried out, a test for APEOs content in tea soup is required. (We consume APEOs by drinking tea soup.) The matrix of tea soup is completely different from that of tea, which requires the establishment of a new validation method. The overall workload has more than doubled, which cannot be completed in a short time.

  1. Preparing references is not in accordance with the requirements of the journal. Titles of articles should be abbreviated and capitalized: Author 1, A.B.; Author 2, C.D. Title of the article. Abbreviated Journal NameYearVolume, page range.

Response:

Thank you for pointing that out. We have carefully revised the references according to the requirements of the journal. (References, page 11-13)

  1. Please, corrected the word „china” line 106 page 15; the abbreviation “quechers” line 71 page 17 Please, check all position in references very carefully.

Response:

Thank you for pointing that out. We have carefully revised them. (Line 341, page 12; Line 395, page 13)
